# Fatigue Properties of Maraging Steel after Laser Peening

**Luca Petan [1], Janez Grum [1,*], Juan Antonio Porro [2], José Luis Ocaña [2]**  **and Roman Šturm [1]**

[1] Faculty of Mechanical Engineering, University of Ljubljana, 1000 Ljubljana, Slovenia; luca.petan@gmail.com (L.P.); roman.sturm@fs.uni-lj.si (R.Š.)

[2] UPM Laser Centre, Polytechnical University of Madrid, 28031 Madrid, Spain; japorro@etsii.upm.es (J.A.P.); jlocana@etsii.upm.es (J.L.O.)

\* Correspondence: janez.grum@fs.uni-lj.si; Tel.: +386-41-725-520

**Abstract:** Maraging steels are precipitation hardening steels used for highly loaded components in aeronautical and tooling industry. They are subjected to thermomechanical loads and wear, which significantly shorten their service life. Improvements of their surface mechanical properties to overcome such phenomena are of great interest. The purpose of our research was to investigate the influence of pulse density and spot size of a laser shock peening (LSP) process on the surface integrity with the fatigue resistance of X2NiCoMo18-9-5 maraging steel. Surface integrity was analyzed through roughness, residual stress, and microhardness measurements. The tests performed on resonant testing machine confirmed LSP is a promising process for increasing fatigue resistance of a component. Fatigue crack occurs, when the resonance frequency decreases. This moment, when the fatigue crack initiation phase ends and the fatigue crack propagation phase starts, was chosen as the moment of failure. We have proved LSP is a successful method in improving fatigue resistance of maraging steel by appropriate combination of laser spot size and pulse density tested in our research.

**Keywords:** microhardness; residual stresses; resonant fatigue resistance; roughness

## 1. Introduction

Ultrahigh-strength maraging steels achieve exceptional mechanical properties through precipitation strengthening [1]. Upon cooling from a solution annealing temperature, a nickel-rich austenite matrix with a virtual absence of carbon transforms to a soft and fully martensitic structure with a high dislocation density [2]. Solution annealing followed by an artificial aging process at a temperature of around 480 °C [3] involves the precipitation of intermetallic compounds at dislocation sites, thus contributing to the achievement of an excellent combination of strength and toughness. Commercially available maraging steels can reach yield strengths of over two GPa. Other characteristics of maraging steels are dimensional stability, good machinability and weldability, high fracture toughness, good thermal conductivity, and significantly high resistance to crack propagation and thermal fatigue [4]. Properties, such as good machinability, dimensional stability during heat treatment, and significantly high resistance to thermal fatigue, are needed in tooling applications, such as plastic molds and die casting dies for magnesium and aluminum alloys.

When used as a structural steel, maraging components can be exposed to various forms of detrimental phenomena, such as high-cycle mechanical fatigue [5], synergetic effects of stress and corrosion [6], wear, and thermal fatigue [7], which causes heat checks and stress cracks.

Mechanical properties and fatigue behavior of highly stressed metallic components can be significantly improved by generating compressive residual stresses (RSs) in the surface layer of a material using peening techniques [8], such as shot peening (SP) and laser shock peening (LSP). LSP can

produce highly compressive RSs up to 1 mm in depth, which is about four times deeper than with a traditional SP process [9]. LSP is an innovative surface treatment [10], during which the surface of a treated component, usually covered with an absorbent coating and a transparent confining medium, is exposed to nanosecond long laser pulses of intense energy [11–13]. The irradiated zone vaporizes and transforms into plasma by ionization. Rapid expansion of the high-temperature plasma generates pressure, which is transmitted into the metal through shock waves. The movement of the shock waves from the surface to the depth of the material causes in-plane expansion of the material. When the stresses, caused by the shock waves, exceed the dynamic yield strength of the material, plastic deformation occurs. Those changes in the material generate compressive RSs.

A lot of research work was done on conventional steels, but very little is known about the effects of LSP on the fatigue strength of maraging steels. Banas et al. [14], who exposed maraging steel weldments to high-power Nd:YAG laser pulses, presented one of the early research papers in this context. The mechanical effect of shock waves increased the dislocation density in a heat-affected zone (HAZ) that led to a 17% increase in fatigue strength after the LSP. Grum et al. [15] analyzed the effects of LSP on a die casting maraging steel, i.e., X2NiCoMo12-8-8. They found out the LSP generated highly compressive RSs in the component surface layer. Petan et al. [16] have also shown that LSP with relatively low pulse energy can generate compressive RSs in maraging steel at a level of 500 MPa. When they increased laser pulse density (PD), the surface roughness increased. Similarly, Lavender et al. [17] found out compressive RSs in pilger dies, made of A2 tool steel, were produced by the effects of LSP. The RSs reached a depth of 1.5 mm with the maximum surface values up to −1050 MPa. This influenced an increase of the fatigue life of the pilger dies by 300%. Studies [18–21] on other high-strength tool steels have shown that the implementation of peening techniques can increase wear and thermal fatigue resistance by generating compressive RSs and inducing strain hardening in a surface layer.

The purpose of our research was to investigate the influence of laser processing parameters on the effects of the surface properties with fatigue resistance of X2NiCoMo18-9-5 maraging steel. Surface integrity was analyzed with surface roughness, residual stress, microhardness measurements, and resonant fatigue tests, while the influence of each processing parameter and their interactions was statistically evaluated using the analysis of variance (ANOVA) [22].

## 2. Materials and Methods

### 2.1. Material Properties

Experimental work was conducted on the 300-grade X2NiCoMo18-9-5 maraging steel, which was provided by Deutsche Edelstahlwerke. The given chemical composition of the maraging steel is presented in Table 1. Specimens prepared for fatigue tests were cut out of a delivered maraging rolled plate with a thickness of 9.5 mm (the width of the fatigue test specimen) and then heat-treated. The specimens were first solution annealed for 1 h at a temperature of 820 °C, followed by quenching, and artificially ageing for 3 h at 480 °C, cooled in air (this process was called maraging precipitation hardening (MPH)). The mechanical properties of the precipitation-hardened maraging steel are listed in Table 2. Before performing the LSP, all the specimens were ground and polished to ensure surface roughness uniformity.

**Table 1.** Nominal chemical composition of the X2NiCoMo18-9-5 maraging steel (unit in wt. %).

| Fe | Ni | Co | Mo | Ti | Al | C |
|------|-------|------|---------|---------|-----------|-------|
| Bal. | 17–19 | 8–10 | 4.5–5.5 | 0.5–0.8 | 0.05–0.15 | ≤0.03 |

**Table 2.** Mechanical properties of the X2NiCoMo18-9-5 maraging steel after the maraging precipitation hardening (MPH) process.

| Tensile Strength [MPa] | Young's Modulus [GPa] | Density [kg/m$^3$] | Elongation in 50 mm [%] | Reduction in Area [%] | Fracture Toughness [MPa·m$^{1/2}$] | Rockwell Hardness [Hardness Rockwell Scale C (HRC)] |
|---|---|---|---|---|---|---|
| 1800–2100 | 195 | 8100 | 8–9 | 40–53 | 67–80 | 52–56 |

## 2.2. Laser Shock Peening

The heat-treated and polished specimens of maraging steel (l = 80 mm × w = 20 mm × t = 9.5 mm) were exposed to LSP in a confined mode using water as a transparent overlay and without an absorbent coating. In comparison to the conventional LSP with an absorbent coating, this technique uses lower laser pulse energy in order to avoid surface melting. This permits laser treatment to be performed without a sacrificial layer and allows for higher overlapping rates between laser spots, in our case between 78% and 92%. The LSP path mode on the specimens is presented in Figure 1. The direction of the laser processing was parallel to the rolling direction. Several specimens were exposed to LSP simultaneously. The width of the LSP area on each specimen was 16 mm. We used different combinations of laser pulse parameters. We changed the laser PD in a range from 900 to 2500 cm$^{-2}$ with a step of 100 cm$^{-2}$ at three different laser spot diameters (SDs; i.e., 1.5, 2.0, and 2.5 mm). The LSP processing parameters used in the experiment are presented in Table 3. The Laser pulse energy and the laser pulse duration were constant, i.e., 2.8 J and 10 ns, respectively. Therefore, the laser hit the specimen surface with a power density in a range between 5.7 and 15.8 GW·cm$^{-2}$. We used a Q-switched Nd: YAG laser (model: Brilliant B, wavelength: 1064 nm, Gaussian spatial distribution; Quantel, Lannion, France) operating at 10 Hz.

**Table 3.** Research design: laser shock peening (LSP) process parameters and plan of measurements.

| Laser Spot Diameter (SD) [mm] | Laser Power Density [GW·cm$^{-2}$] | Laser Pulse Density (PD) [cm$^{-2}$] | Pulse Overlapping Rate [%] | Measurements of Fatigue Resistance |
|---|---|---|---|---|
| 1.5 | 15.8 | 900, 1600, and 2500 | 78–87 | Surface roughness |
| 2.0 | 8.9 | 900, 1600, and 2500 | 83–90 | Hardness |
| 2.5 | 5.7 | 900, 1600, and 2500 | 87–92 | Residual stresses |

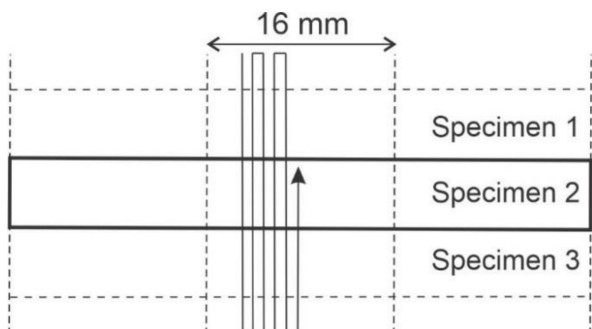

**Figure 1.** Principle of the LSP path mode on the specimens.

## 2.3. Resonant Fatigue Tests

In order to obtain insight into the effects of LSP on the mechanical fatigue resistance of the chosen maraging steel, fatigue tests were carried out using a resonant testing machine, CRACKTRONIC (RUMUL, Neuhausen am Rheinfall, The Switzerland), which is a table model for dynamic bending load applications with testing frequencies between 40 and 300 Hz. The kinematic conditions allows for

pure bending between the gripping heads. An electromagnetically driven resonator, built as a rotary oscillator, creates appropriate bending moments. The elastic modulus of a material and the specimen geometry have effects on the resonance frequency of the machine. The crack initiation and propagation in this unit reduce its cross-sectional area, which affects stiffness reduction and consequently resonance frequency, as stated also in [23]. This machine measures resonance frequencies with a resolution of 0.01 Hz. In this research, the bending moment $M$ was applied in a sinusoidal wave form at a stress ratio $R$ of 0.1. We chose this stress ratio to provide and keep the upper surface of the specimen, where we expected crack initiation, in tensile conditions during fatigue tests. The bending frequency was around 114 Hz. We further reduced the middle of the specimen to ensure higher semicircular bending stresses in this area (Figure 2). In our resonant tests, we applied bending moments in a range between 60 and 78 N·m. The parameters of fatigue loading used in the experiment are presented in Table 4. According to our simulation, the maximum bending moment within a fatigue cycle generated a load of 1082 MPa in the most critical point of the specimen. The stress map during the bending is represented in Figure 3. Stress analysis was simulated with a software package SolidWorks (v23, 2015) according to the finite elements method (FEM) (3DEXPERIENCE, Vélizy-Villacoublay, France).

**Table 4.** Fatigue loading parameters.

| Laser SD [mm] | Laser PD [cm$^{-2}$] | Maximum Bending Moment [N·m] | Maximum Bending Stress [MPa] |
|---|---|---|---|
| 1.5 | 900 and 2500 | 60, 66, 72, and 78 | 833, 916, 999, and 1082 |
| 2.0 | 1600 | 60, 66, 72, and 78 | 833, 916, 999, and 1082 |
| 2.5 | 900 and 2500 | 60 and 78 | 833 and 1082 |
| Base metal (MPH) | - | 60, 66, 72, and 78 | 833, 916, 999, and 1082 |

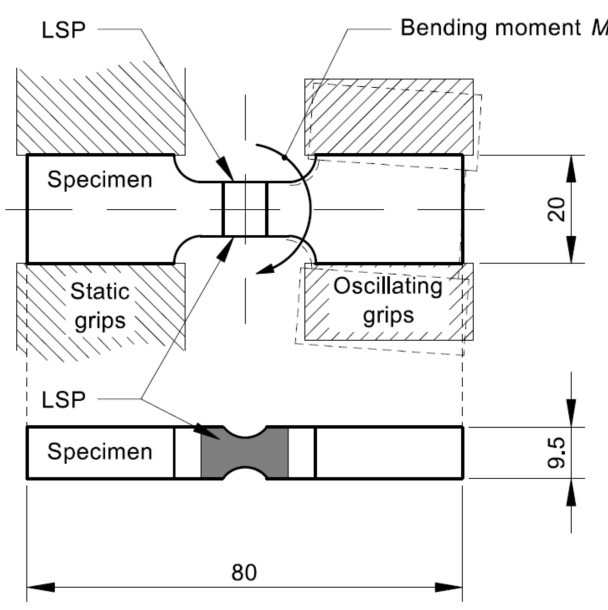

**Figure 2.** Principle of mechanical fatigue testing. The bending moment ($M$) range: 60–78 N·m.

Figure 4 shows the change of the resonant frequency in dependence of the number of fatigue cycles, where $N_i$ represents an initiation period and $N_p$ indicates a propagation period. The resonant frequency, conditioned by the specimen's geometry, began to decrease, when the fatigue crack occurred. This event, which also separated the fatigue crack initiation phase and the fatigue crack propagation phase, was chosen as the moment of failure. We did not measure specimens' temperatures during the fatigue testing, and we did not notice any temperature increase in the specimens during and after the

test. The CRACKTRONIC is a compact testing device, so heat generated in the specimen during the fatigue loading could possibly be transferred to the resonant device.

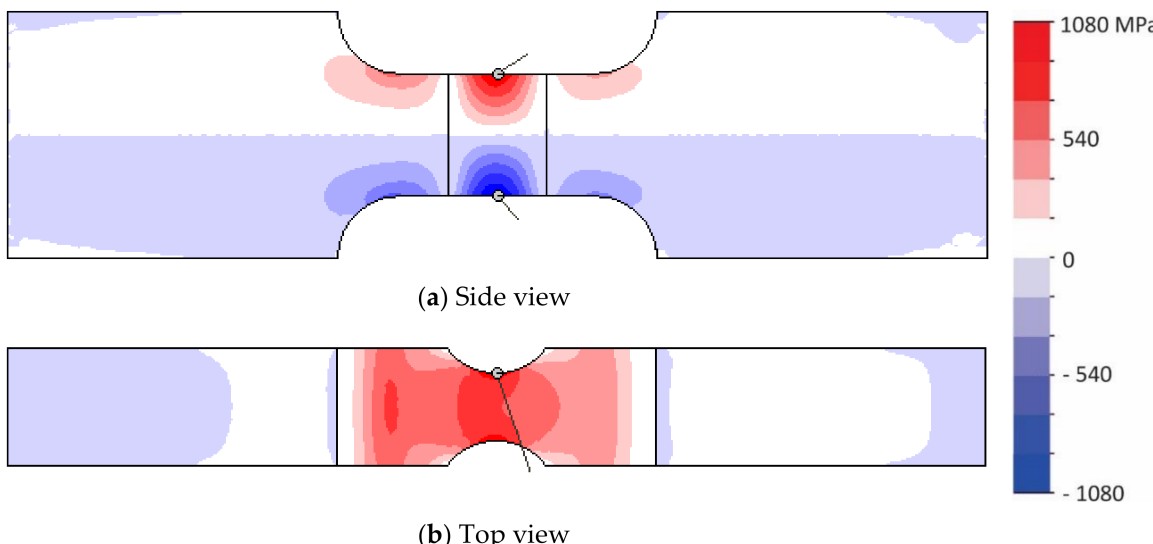

(**a**) Side view

(**b**) Top view

**Figure 3.** Stress maps during the bending within the fatigue specimen: (**a**) side-view stress map; (**b**) top-view stress map. The maximum stress indicated in red was approximately 1000 MPa. The bending moment was 78 N·m.

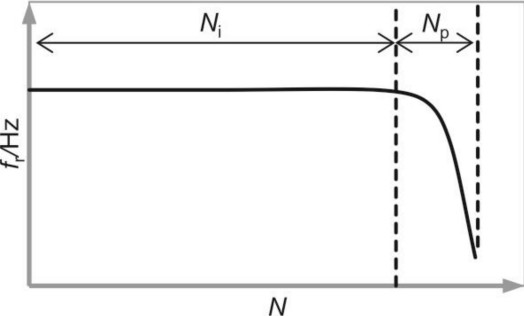

**Figure 4.** Resonant frequency as a function of the number of fatigue cycles during the fatigue test.

## 3. Results and Discussion

### 3.1. Roughness Measurements

Surface roughness measurements ($Ra$—the arithmetic average of the absolute values of the profile heights over the evaluation length, and $Rz$—the average value of the absolute values of the heights of five highest-profile peaks and the depths of five deepest valleys within the evaluation length) were performed with a Surtronic 3+ contact profilometer (Taylor Hobson Ltd, Leicester, UK). Before the LSP was applied, 10 individual surface roughness measurements were carried out, with five in the rolling direction and five in the transversal direction. The number of measurements after the LSP was doubled. The characteristic surface roughness was calculated as an average of the results from longitudinal and transversal measurements.

Before the LSP, the surface roughness was at a level of $Ra$ = 0.2 μm and $Rz$ = 1.2 μm. After the LSP, the surface roughness increased to $Ra$ = 0.6–1.2 μm and $Rz$ = 3.9–7.4 μm. The minimum values of roughness were measured on the specimen treated with the smallest laser spots and the lowest PD, while the maximum values were measured on the specimen treated with the largest laser spots and the highest PD.

To obtain an overview of the effects of laser PD and laser spot size on the surface integrity evaluated by roughness, hardness, and RS, a general factorial design was carried out. The laser parameters were examined using the ANOVA and the response surface methodology (RSM), where the influence of individual factor was considered to be statistically significant for $P < 0.05$. According to the statistical analysis (ANOVA), we found out that interactions of PD and SD also had a significant influence on surface roughness, RS, and hardness ($P < 0.0001$). We tested several polynomial models on statistical characteristics (*F*-value (statistical characteristics used to test the significance of adding new model terms to those terms already in the model), $R^2$ (reports the strength of the relationship between the set of independent variables and the dependent variable)) to fit the response to the measured values. We found out, according to the *F*-values and the $R^2$ values, the most suitable model for surface roughness, hardness, and RS is a quadratic model. Therefore, according to the mentioned measurements at laser SDs of 1.5, 2.0, and 2.5 mm, the results are presented as contour plots.

As can be observed in the contour plots in Figures 5 and 6, laser PD had the main influence on surface roughness. Higher PD influenced higher surface roughness, at all the diameters of the laser spot. When processing hard materials, it is difficult to detect a clear relation between LSP processing parameters and surface roughness due to moderate increases of *Ra* and *Rz*. At a PD of 900 cm$^{-2}$, the minimum roughness values were obtained when using an SD of 1.9 mm. At a PD of 1600 cm$^{-2}$, the minimum roughness values were obtained when using an SD of 1.7 mm. At a PD of 2500 cm$^{-2}$, the minimum roughness values were obtained when using an SD of 1.5 mm. That happened partially at the expense of decreasing the overlapping of laser spots during the LSP. At the highest PD, we can reduce surface roughness by decreasing laser spot size. At a lower PD, the optimal laser SD was between 1.8 and 2.0 mm.

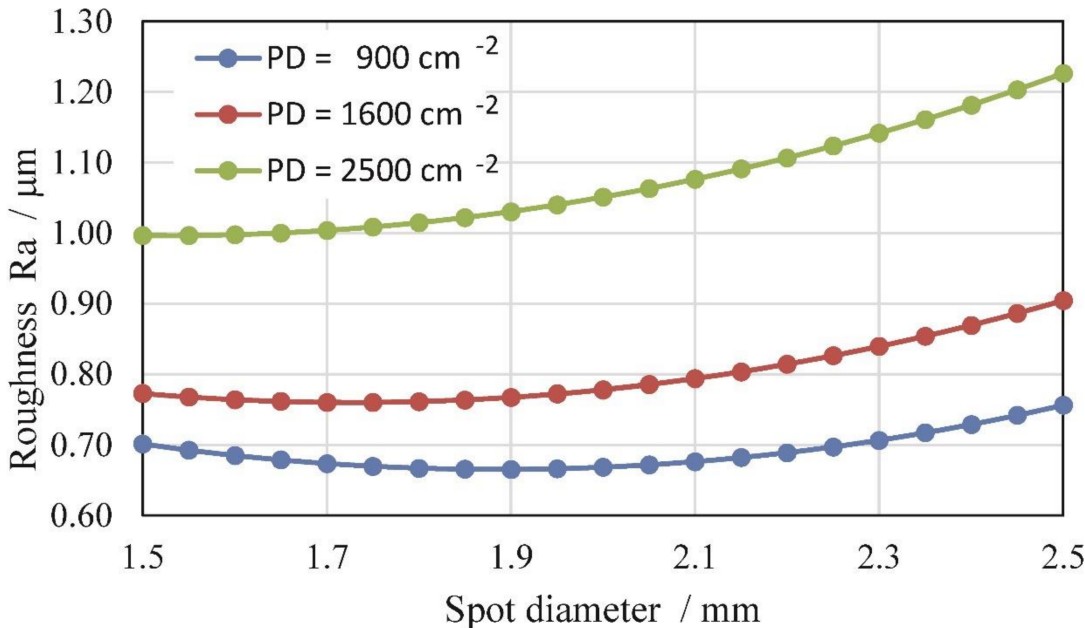

**Figure 5.** Roughness (*Ra*) as a function of laser SD and PD.

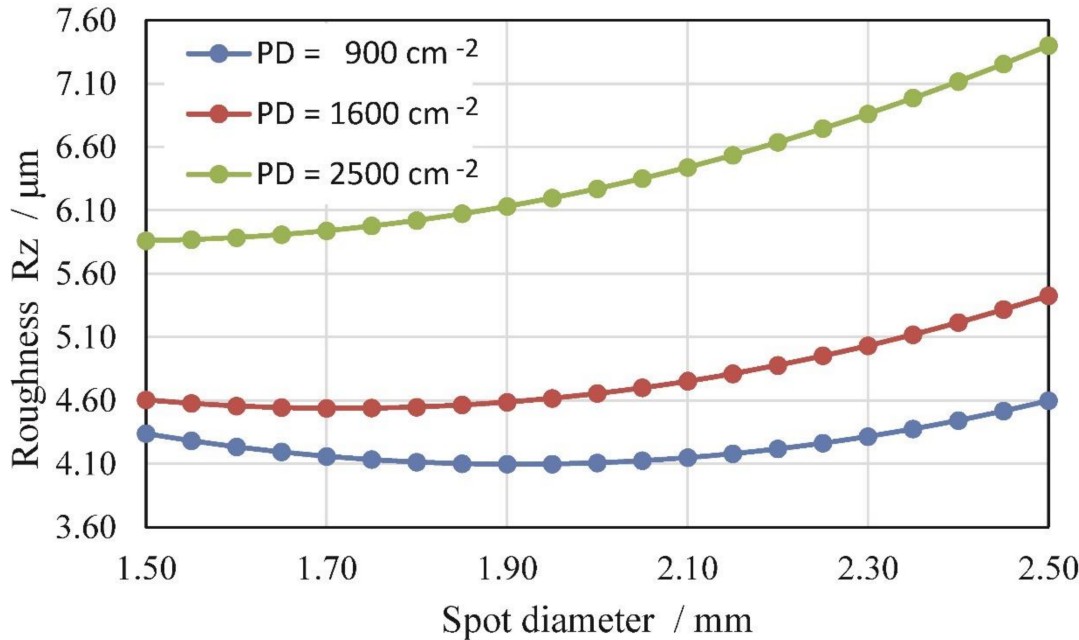

**Figure 6.** Roughness (*Rz*) as a function of laser SD and PD.

The ablative nature of a laser process, because of the absence of an absorbent coating, combined with mechanical effects of laser pulse pressure, leads to profile deepening.

Profile depth (Pt) is a kind of important information for planning additional process operations after LSP, like grinding and polishing. It shows us the height difference between the untreated and LSP-treated surfaces (inserted in Figure 7). We can find out that increasing PD influences the increase of profile depth and it is more distinct at a large diameter of a laser spot. Profile depth increases linearly in conjunction with increasing PD. The line is tilt more greatly in the case of bigger SDs. Especially the trend of Pt results is very unfavorable when applying LSP with a laser SD of 2.5 mm (Figure 7). The surface profile was lowered by almost 100 µm in the case of PD = 2500 cm$^{-2}$.

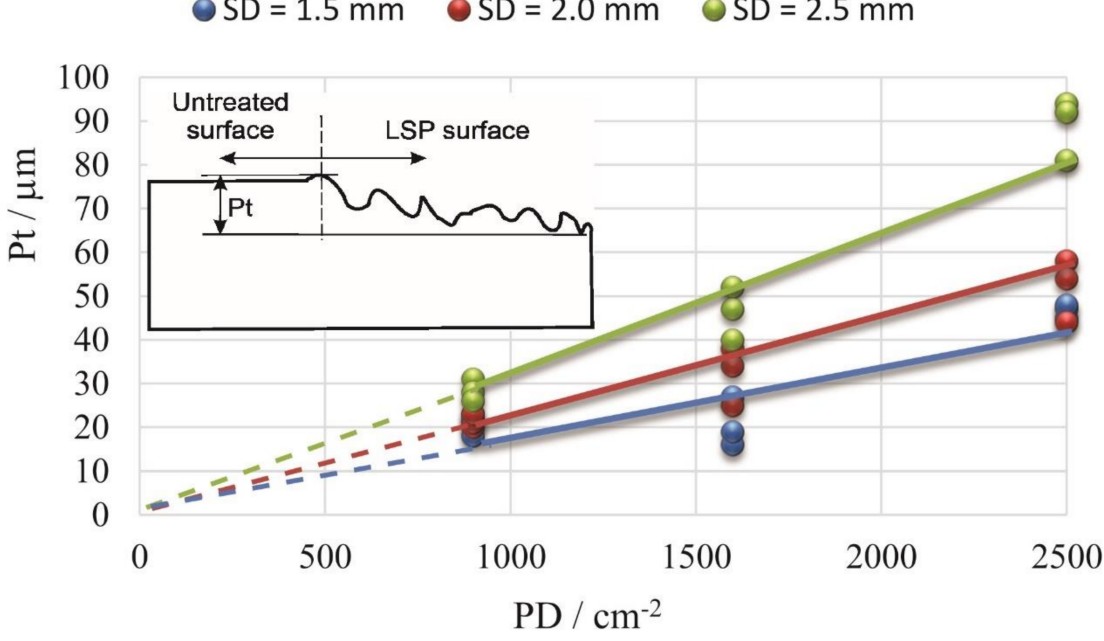

**Figure 7.** Profile depth (Pt) as a function of laser PD and SD.

### 3.2. Residual Stress Measurements

RSs were measured with a standard hole-drilling method. We used a milling cutter with a diameter of 1.6 mm. Deformation of the specimen caused during the drilling was measured with 06-062-UM strain gage rosettes, which were connected to the LabVIEW software. Blind holes in the surface layer were incrementally drilled to a depth of 1 mm with a 0.1 mm increment, where the drilling process was temporarily interrupted, enabling deformations resulting from material relaxation to fully occur and stabilize. The final RS profiles were calculated with the H-Drill software, where an integral method with automatic smoothing was applied.

In the base metal, after the heat treatment, RSs were low, ranging between −13 and 49 MPa. After the LSP, at different process parameters, compressive RSs arose in the surface layer. The maximum compressive RSs were at the surface in a range between −1000 and −350 MPa. At depths between 0.5 and 1.0 mm, the transition from a compressive state to a tensile state occurred. Some typical RS distributions in depth are presented in Figure 8. According to the statistical analysis (ANOVA), we found out that the interactions of PD and SD have a significant influence on the RS ($P < 0.0001$). We fitted the measured values with the quadratic model. Therefore, according to the mentioned measurements at laser SDs of 1.5, 2.0, and 2.5 mm and at laser PDs of 900, 1600, and 2500 cm$^{-2}$, the results are presented as contour plots for the whole range of laser SD from 1.5 to 2.5 mm.

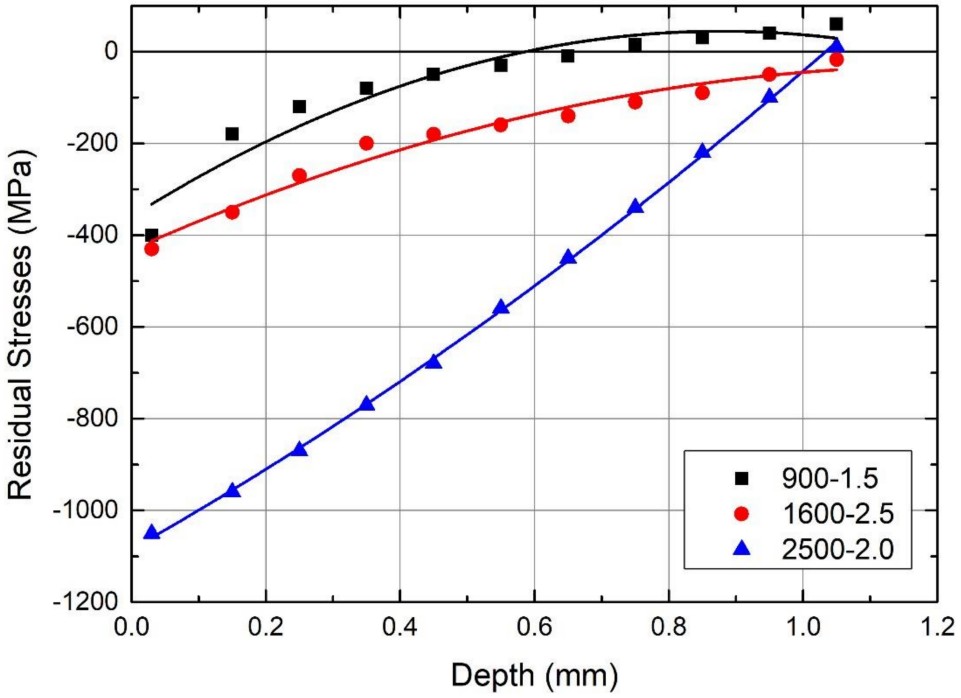

**Figure 8.** Residual stress distributions in depth for different PDs (i.e., 900, 1600, and 2500 cm$^{-2}$) and different SDs (i.e., 1.5, 2.0, and 2.5 mm).

The plots of the average RSs at specified depths, shown in Figures 9 and 10, suggest that the maximum compressive RSs were achieved with a 2.0 mm-diameter laser spot, both at the surface and at a depth of 1.0 mm. This could not be directly connected with the pulse power density (PPD) only, which is the highest with a 1.5 mm-diameter laser spot. The interaction between laser beam size and material affected the propagation nature of the shock waves. Smaller-diameter shock waves probably expand like spheres with a higher attenuation rate than larger-diameter shock waves, which behave like planar fronts [15,16,24,25]. This phenomenon, together with a higher overlapping rate between laser spots, may explain our findings. The RS profiles are almost the same for different PDs at a depth of 1.0 mm. The shift from big compressive RSs at the surface towards tensile RSs at a depth of 1.0 mm

is between 400 and 600 MPa. In Figure 11, we can see a comparison of the RS profiles at the surface and at a depth of 1.0 mm for PD = 1600 cm$^{-2}$.

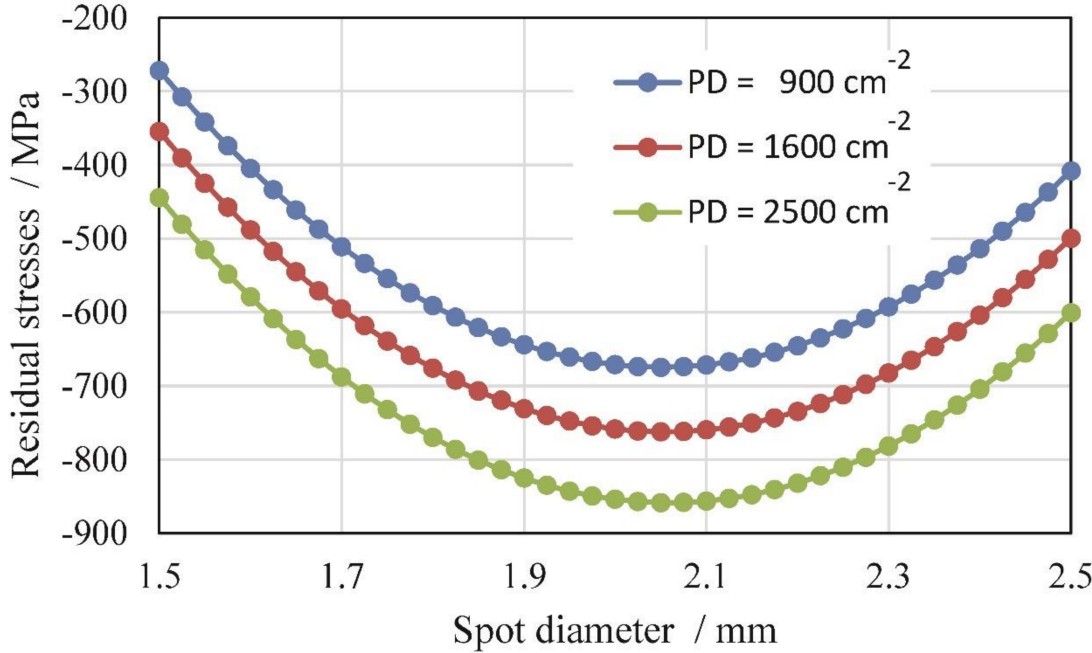

**Figure 9.** Residual stresses at the surface as a function of laser SD and PD.

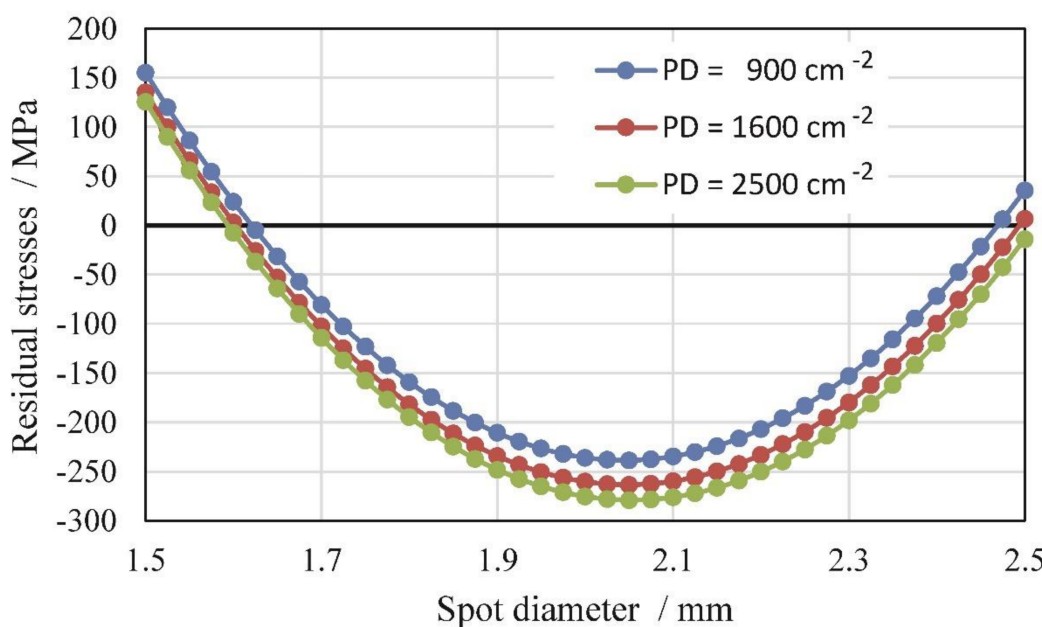

**Figure 10.** Residual stresses at a depth of 1.0 mm as a function of laser SD and PD.

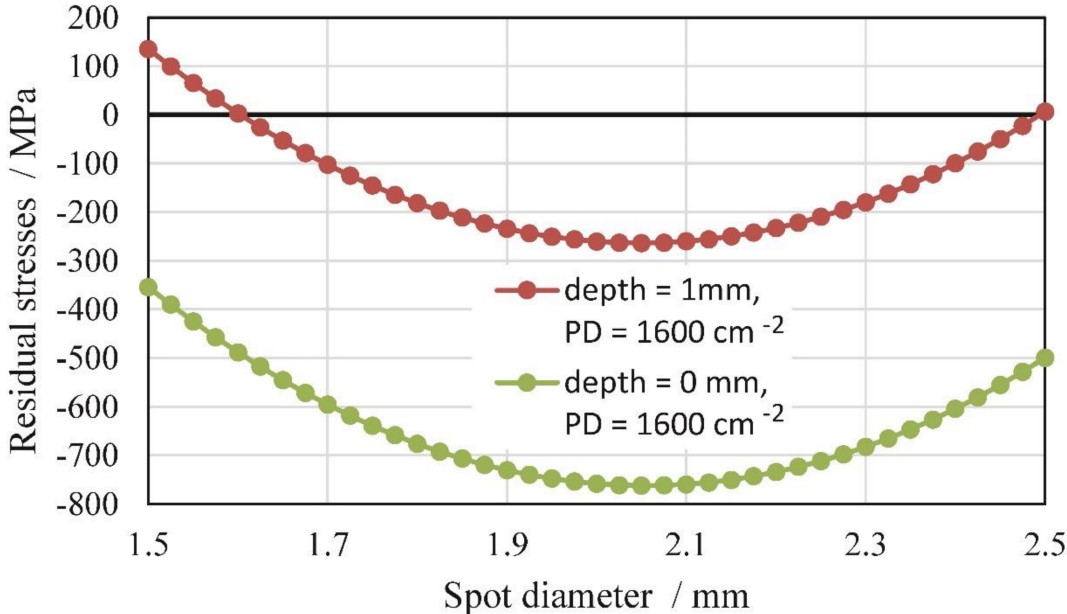

**Figure 11.** Residual stresses at the surface and at a depth of 1.0 mm as a function of laser SD for PD = 1600 cm$^{-2}$.

### 3.3. Microhardness Measurements

In-depth microhardness distributions were obtained with the standard Vickers test method using a Leitz Wetzlar hardness tester with a 200 g load and a 15 s load time. Vickers microhardness was measured at a depth of 2 mm from the LSP-treated surface (Figure 12). The microhardness analysis of the surface layer indicated that, after the LSP, strain hardening occurred. Strain hardening was detected as an increase in microhardness. The microhardness of the heat-treated maraging steel before the LSP was 667 $HV_{0.2}$. The highest microhardness value after the LSP was measured just below the surface in a range between 730 and 740 $HV_{0.2}$. The increased hardness was detected in the surface layer. The thickness of the strain-hardened surface was between 0.6 and 1.6 mm, depending on laser PD and laser SD.

Interactions of laser PD and SD have a significant influence on the hardness of an LSP-treated surface, found by a statistical analysis (ANOVA) ($P < 0.0001$) [25]. We fitted the measured values with the quadratic model. The results are presented as contour plots for the whole range of laser SD from 1.5 to 2.5 mm. The contour plots, shown in Figures 13 and 14, indicate that, in general, the maximum microhardness was achieved with a 2.0 mm-diameter laser spot, both at the surface and at a depth of 1.0 mm. Once again, this could not be directly connected with PPD. In the case of a 1.5 mm-diameter laser spot, the PPD was higher than in the case of a 2.0 or 2.5 mm-diameter laser spot. These findings confirm that, for our laser source, with a 2.0 mm-diameter laser spot, the most pronounced mechanical effect was obtained.

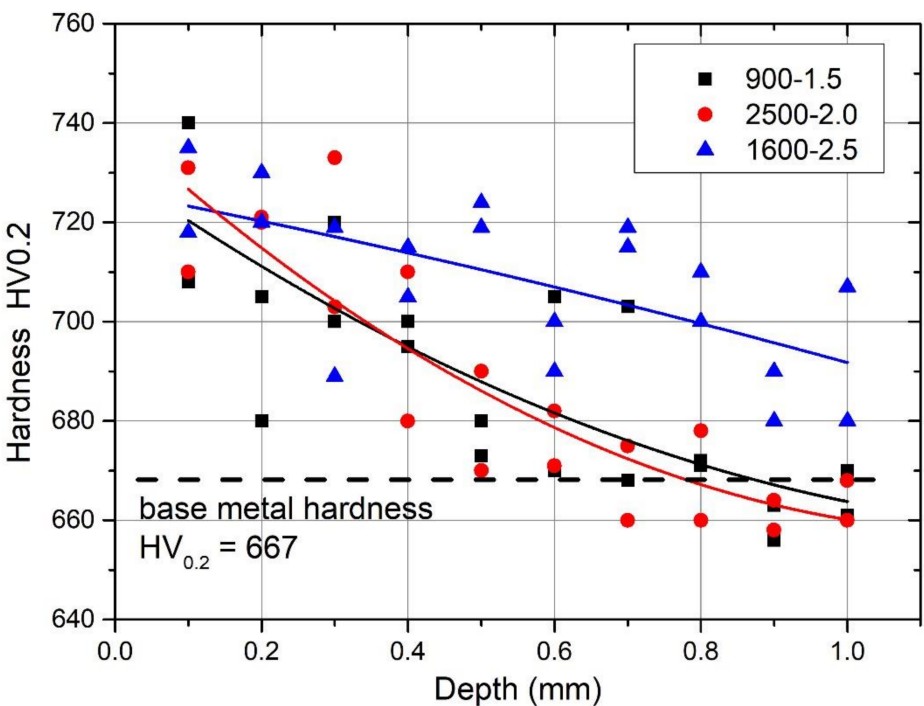

**Figure 12.** Micro-hardness measurements in depth for different PDs (i.e., 900, 1600, and 2500 cm$^{-2}$) and different SDs (i.e., 1.5, 2.0, and 2.5 mm).

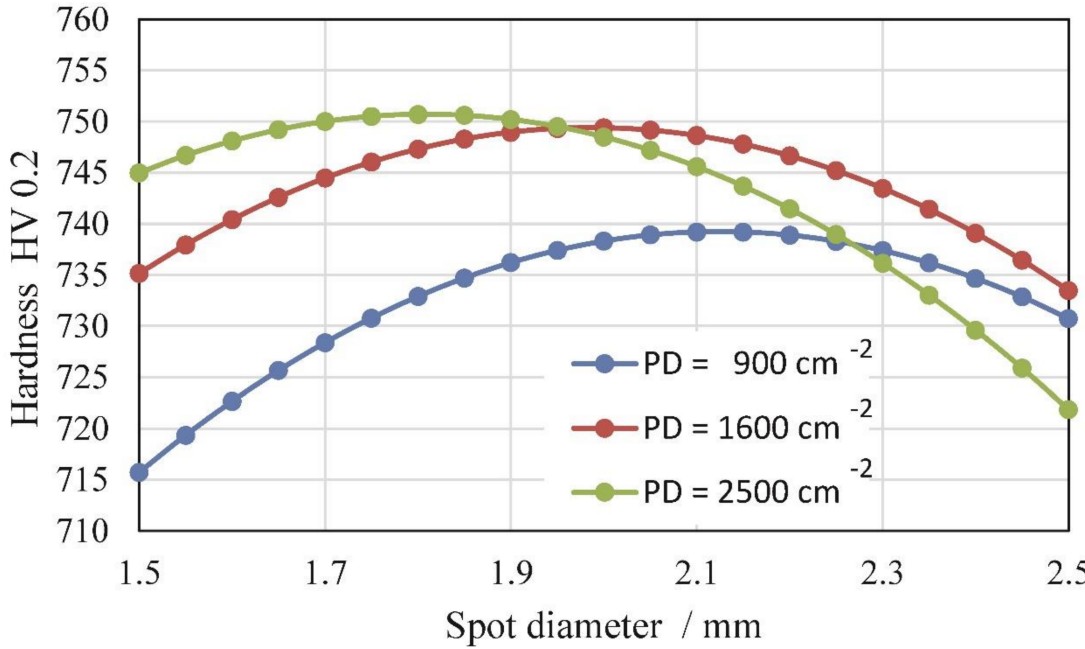

**Figure 13.** Microhardness at the surface as a function of laser SD for different PDs (i.e., 900, 1600, and 2500 cm$^{-2}$).

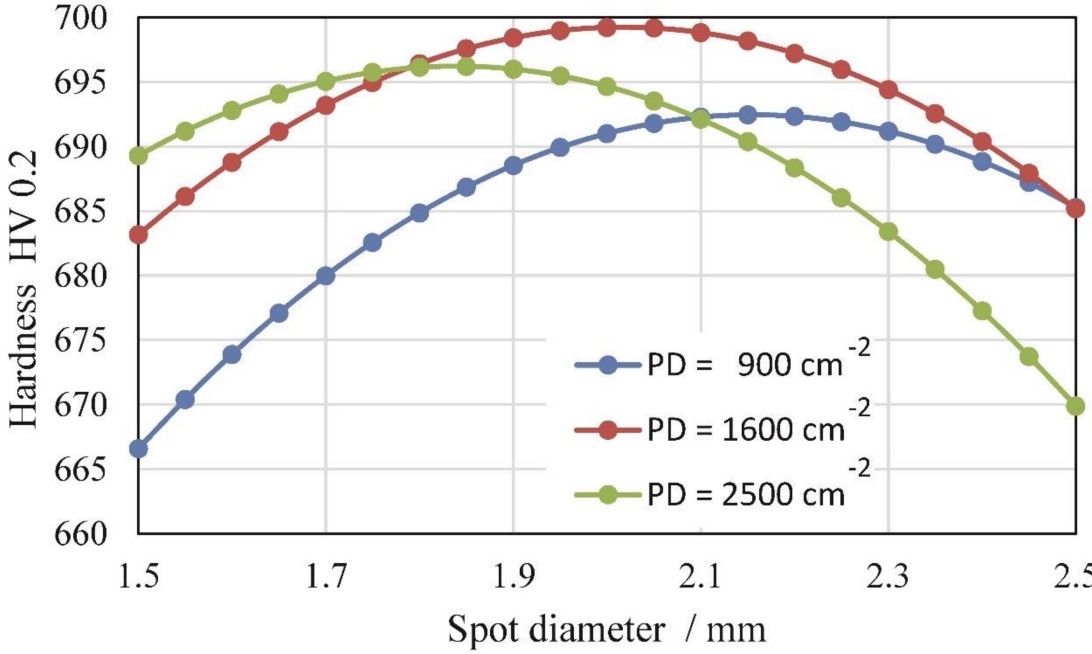

**Figure 14.** Microhardness at a depth of 1.0 mm as a function of laser SD for different PDs (i.e., 900, 1600, and 2500 cm$^{-2}$).

Additionally, other researchers found out that higher overlapping rates and multiple LSP impacts tend to cause a hardness increase [4,9]. We have previously shown that hardness increases after LSP is mainly due to the presence of compressive RSs [15,22]. With different LSP parameters, the microhardness decrease from the surface to a depth of 1.0 mm is almost linear. The relationship between microhardness and laser SD for PD = 1600 cm$^{-2}$ is shown in Figure 15.

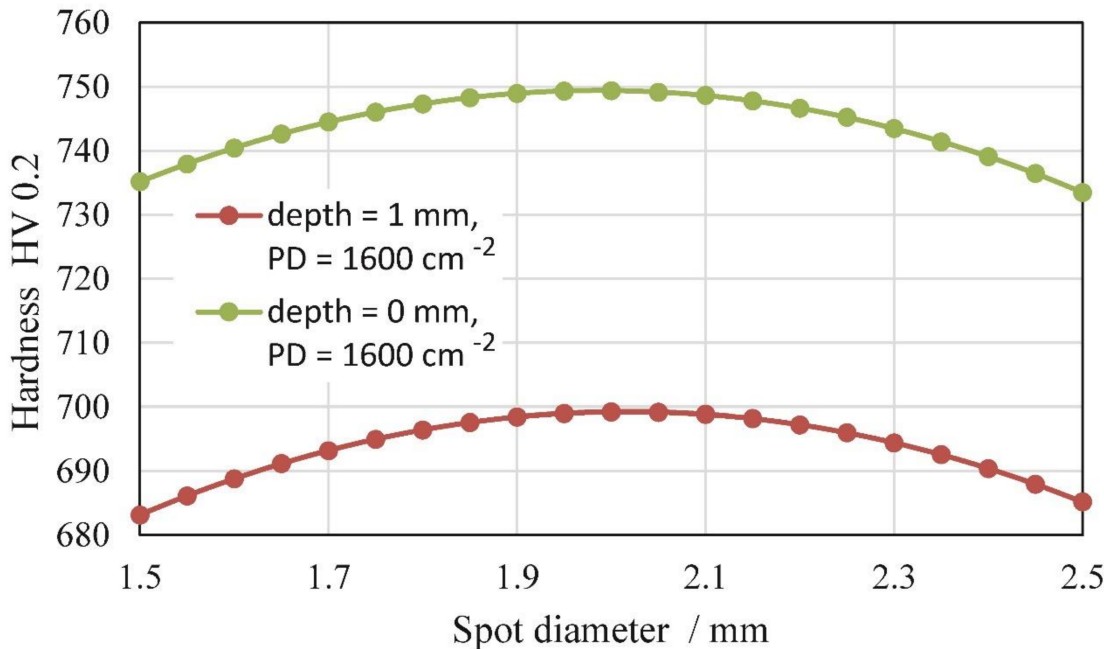

**Figure 15.** Microhardness at the surface and at a depth of 1.0 mm as a function of laser SD for PD = 1600 cm$^{-2}$.

### 3.4. Resonant Fatigue Resistance

The presence of compressive RS itself does not insure an improvement in fatigue strength. The increase in surface roughness must be considered, because surface defects may promote initialization and propagation of cracks during dynamic loading. The fatigue test results are shown in Figure 16. It was found out that the maraging steel in a delivered state (maraging solution annealing, with hardness values of 32–37 hardness Rockwell scale C (HRC) and an *Rm* value of 1200 MPa) withstood the same number of fatigue cycles for crack initiation, which was precipitation-hardened, when the fatigue testing was performed at a bending moment of 60 N·m and a bending stress of 833 Mpa. When the bending stress was increased to 1082 Mpa, the precipitation-hardened maraging steel showed a 43% increase in crack initiation time (blue line in Figure 16). As can be observed, the LSP was proven to be successful in improving the fatigue resistance of precipitation-hardened maraging steel. The fatigue resistance was improved after every combination of laser SD and PD, which was tested within our research. Therefore, the negative effect of the increased surface roughness did not overcome the positive effect of the compressive RSs, present in the thin surface layer of the laser-peened material. According to our statistical analysis, we discovered that the selection of laser SD between 1.5 and 2.5 mm in combination with a PD of 900 or even 2500 cm$^{-2}$ did not have a statistically significant effect on the fatigue resistance when comparing the LSP-treated specimens. The selection of laser processing parameters had a statistically significant effect on fatigue life only, when a laser SD of 2.0 mm was used ($P < 0.0001$). The LSP with an SD of 2.0 mm and a PD of 1600 cm$^{-2}$ increased the fatigue life of the precipitation-hardened maraging steel specimens from a range of 2–4 $\times 10^4$ cycles to a range of 5–9 $\times 10^5$ cycles. Therefore, the number of fatigue cycles, necessary for fatigue crack initiation, increased by 25 times. In some cases, we stopped the fatigue test after $10^7$ cycles without crack initiation.

The implementation of the LSP technique can increase fatigue resistance, by generating compressive RSs and inducing strain hardening in most loaded surface areas, which consequently reduces the need for tool repair. Hence, when we succeed in increasing tool maintenance intervals (repair or change), the production cost is reduced.

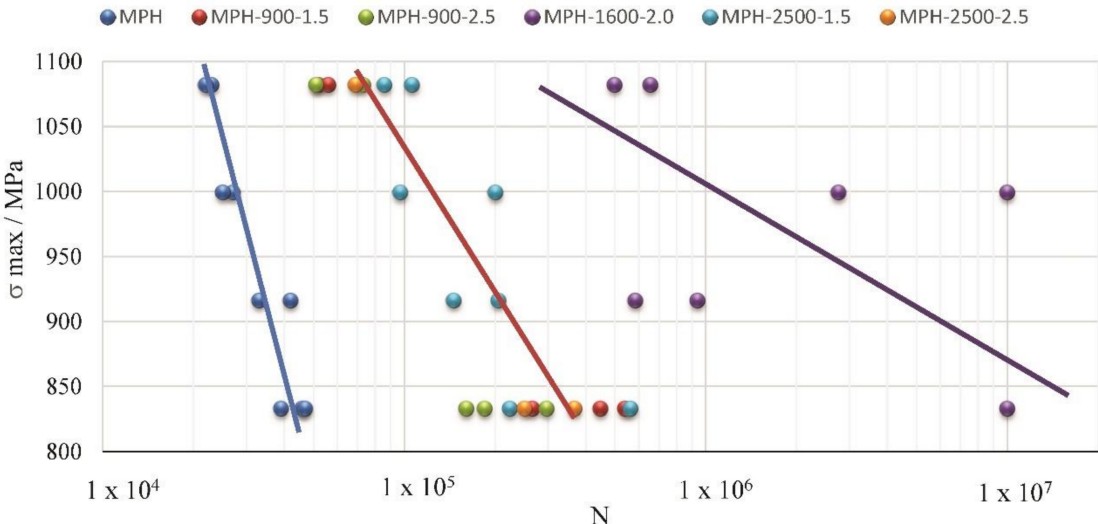

**Figure 16.** Fatigue test results.

Further, the cyclic loading after the crack initiation phase causes crack propagation and leads to a continuous decrease of the resonant frequency until final fracture. Other researchers have also studied the effect of LSP on crack propagation behavior [24]. Within our research, we compared the behavior of the resonant frequency between the untreated and laser-peened specimens during the fatigue loading.

As can be observed in Figure 17, the resonant frequency was decreasing more slowly, when the surface was treated with LSP. This finding indicated that laser peening not only extended the fatigue crack initiation time, but also reduced the crack propagation rate.

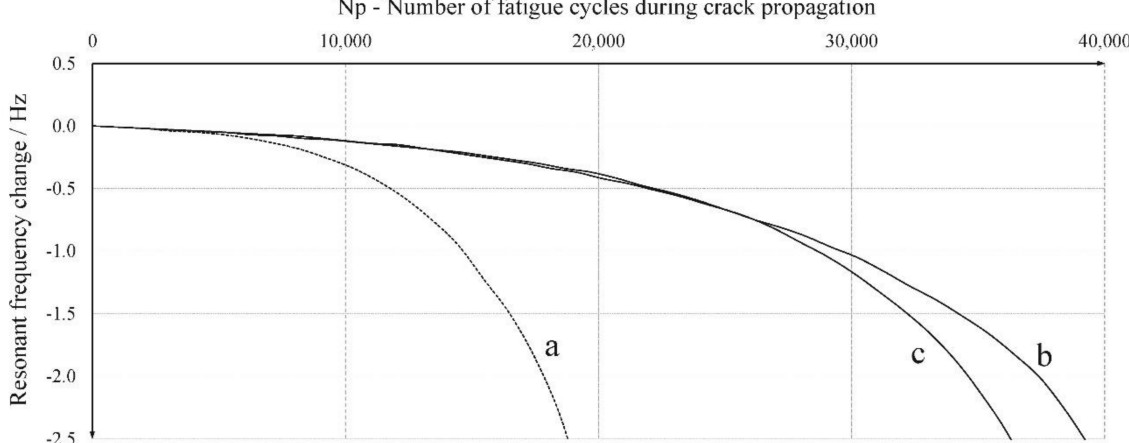

**Figure 17.** Resonant frequency behavior within the crack propagation phase during the fatigue bending: (**a**) MPH; (**b**) MPH-900-2.5; and (**c**) MPH-1600-2.0. The maximum bending moment was 78 N·m.

Figure 18a presents the failure locations on the specimens without LSP. The crack grew at the surface, because the LSP process slowed down thanks to the presence of the compressive RSs.

By laser peening with a 2.0 mm-diameter laser spot, the generated compressive RSs were in a range of 800–900 MPa, such that their positive effect could overcome the critical stress on the narrowed site in the middle of the fatigue specimen. It can be observed that the fatigue crack initiated at the edge of the LSP area (Figure 18c) and was not in the narrowest position in the middle of the specimen (Figure 18b).

When choosing the best laser peening parameters for increasing the fatigue life of a component, we must take into account a combination of the highest hardness and the highest compressive RS at a great depth (in our case, it is 1 mm). Laser spot size, in combination with laser power density and overlapping degree of laser pulses, affects the propagation of shock waves [26]. Small-diameter shock waves have different attenuation rates from those of large-diameter shock waves, when they go deep. In our case, we obtained the best combination of surface quality and material properties with the following LSP parameters: laser PPD = 8.9 GW·cm$^{-2}$, laser PD = 1600 cm$^{-2}$, laser SD = 2 mm, laser spot overlapping degree = 87%.

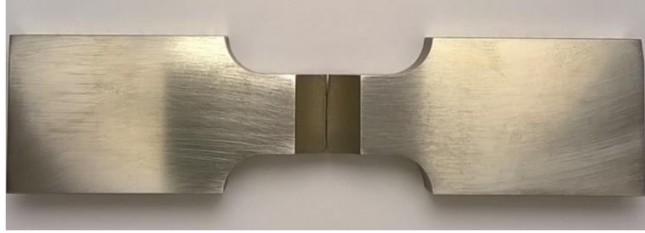

(**a**) Without LSP

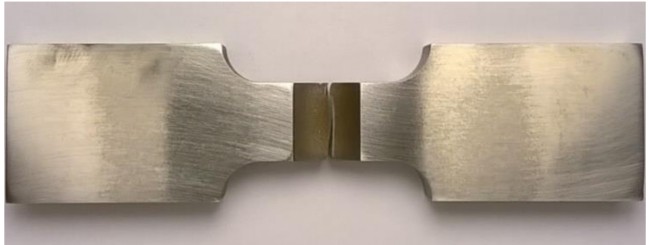

(**b**) With LSP, PD = 900 cm⁻², SD = 2.5 mm, crack in the middle of the specimen

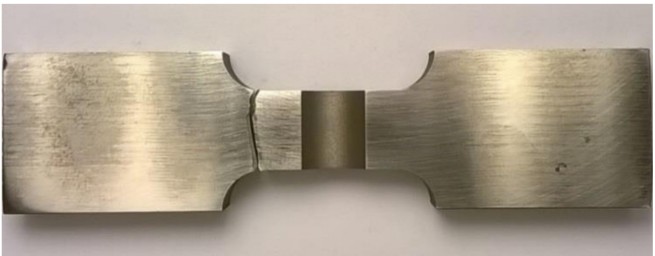

(**c**) With LSP, PD = 1600 cm⁻², SD = 2.0 mm, crack at the edge of LSP area

**Figure 18.** Fatigue crack and failure location on the specimen without LSP (**a**), the specimen with LSP for PD = 900 $cm^{-2}$ and SD = 2.5 mm (**b**), and the specimen with LSP for PD = 1600 $cm^{-2}$ and SD = 2.0 mm (**c**).

## 4. Conclusions

In our research, we have investigated the effects of LSP treatment on the surface properties of the maraging steel X2NiCoMo18-9-5 in a precipitation-hardened state. We have used different combinations of PD and spot size on the surface. The LSP was performed in a confined mode without a protective coating using constant laser pulse energy and duration. The conclusions could be summarized as following:

- After the LSP treatment, the maximum compressive RS of 1050 MPa was generated at the surface. At a depth of 1.0 mm, they still could be at a level of −300 MPa when using the optimal LSP parameters. Otherwise, the transition from a compressive state to a tensile state occurred at depths between 0.5 and 1.0 mm.
- Before the LSP, the surface roughness was $Ra$ = 0.2 μm and $Rz$ = 1.2 μm. After the LSP, the surface roughness increased to $Ra$ = 0.6–1.2 μm and $Rz$ = 3.9–7.4 μm. Higher roughness was obtained, when we increased the laser SD and PD.
- With increasing PD, profile depth Pt was increased. This phenomenon is more distinct at a large diameter of a laser spot. In the case of SD = 2.5 mm and PD = 2500 $cm^{-2}$, the profile depth was almost 100 μm.
- Strain hardening was detected as an increase in microhardness. The microhardness of the precipitation-hardened maraging steel before the LSP was 667 $HV_{0.2}$. The highest microhardness

value at the surface after the LSP is around 750 $HV_{0.2}$ and was registered on the specimen treated with a 2.0 mm-diameter laser spot and laser PDs of 1600 and of 2500 $cm^{-2}$.

- We have obtained the best combination of mechanical properties of the modified surface layer using a 2.0 mm-diameter laser spot.
- The LSP successfully improved the mechanical fatigue resistance of the maraging steel.
- The negative effect of the increased surface roughness did not overcome the positive effect of compressive RSs generated by LSP.
- Considering the chosen testing parameters, the number of fatigue cycles, necessary for fatigue crack initiation, was increased by 25 times.
- By analyzing the resonant frequency during the fatigue testing, we have discovered that the rate of the decrease of the resonant frequency was lower after the LSP, which indicated that laser peening not only extended the fatigue initiation time, but also reduced the crack propagation rate.

**Author Contributions:** J.G. and J.L.O. conceived, designed and performed the experiments; J.A.P. performed LSP experiments, R.Š. and L.P. performed the microstructural, roughness, fatigue, and RS characterizations; L.P. and J.G. analyzed and evaluated the data; L.P. and R.Š. wrote the manuscript. J.G. supervised the study. All the authors reviewed the final paper.

**Funding:** This research received no external funding.

**Acknowledgments:** This paper is part of the research work within the National Research Program (Nr. P2-0270) financed by the Slovene Ministry of Education, Science and Sport. The authors are very grateful for the financial support.

**Conflicts of Interest:** The authors declare no conflict of interest.

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
