# Peer review of "Fatigue Properties of Maraging Steel after Laser Peening"

_metals, doi:10.3390/met9121271_

Round 1

Reviewer 1 Report

Some general comments: 

The work is very well written. It is very well structured. The results are well presented. The conclusions are in accordance with the results. 

Why the figures 5, 6, 7, 8, 9, 10, 11, 12 and 13 have many points for the spot diameter (SD) axe if the authors have only three different laser spot diameters (1.5 mm, 2.0, 2.5 mm) specimens?

Some line by line comments:

Page 5 Line 147-148 “At power density (PD) of... ” The authors want to say “At pulse density…”?

Page 6 Line 161

The authors don’t explain how they measure the profile depth 

What means “Pt”?

Page 7 Line 170

The authors report that the transition from compressive to tensile state occurs at depths between 0.5 mm and 1.0 mm. 

It would be interesting to know the profile of the residual stresses in depth for the different spot diameters because in the base material the residual stresses are very low and, after LSP treatment, they are very high, especially for PD = 2500 cm-2.

Author Response

Why the figures 5, 6, …have many points…?

The explanation is inserted into a text (line 174-184):

Laser parameters were examined using the analysis of variance (ANOVA) and response surface methodology (RSM), where the influence of individual factor is considered to be statistically significant when P<0.05. According to a statistical analysis ANOVA we found, that interactions of pulse density and spot diameter have a significant influence on the surface roughness, residual stresses, and hardness too (P<0.0001). We have tested several polynomial models on statistical characteristics (F-value, R2) to fit the response to measured values. We have found, according to F-values and R2 values, the most suitable model for the surface roughness, hardness, and residual stresses is the quadratic model. So, according to mentioned measurements at laser spot diameter of 1.5, 2.0 and 2.5 mm, the results are presented as contour plots.

Line 147-148: power density (PD) was corrected to pulse density (PD) in text, where needed.

The graphical explanation of profile depth (Pt) measurement is added into Figure 7.

The authors report that the transition from compressive to tensile state of residual stresses…:

It was added new Figure 8: Residual stress distribution in depth for different pulse densities

Reviewer 2 Report

The manuscript deals with  the study of the influence of laser shock peening process (in terms ot pulse density and spot size of) on fatigue resistance of a maraging steel, the X2NiCoMo18-9-5 by adopting several experimental techniques for surface roughness, for residual stress and hardness measurements. Design of experiment was made and the analysis of variance supported in understanding which was the most influent process parameter for fatigue life assessment.

Introduction describes almost well all the research framework, but I found in the manuscript some points to be discussed/clarified/integrated:

Please discuss the cost-effectiveness of the technological process associated with fatigue characterisation and provide some clarification about the novelty of the research and its impact on scientific scenario. Table 2: please include a proper statistics to the data presented. Did you perform tensile tests? Please include a table reporting the research design in terms of process parameters, measurements etc…. Please provide an image of the setup and equipment. Please provide a table resuming fatigue tests: in terms of fatigue loading chosen and applied. Why did you chose R=0.1? Did you observe any temperature increase with the loading frequency?please provide explanations. Please include unit measure in fig.2 A wider discussion on numerical simulation for bending fatigue of material has to be provided. Referring to fig. 8-9 you can estimate the maximum compressive residual stress. I suggest to evaluate it quantitatively and to correlate it with process parameter. This can support the readiness of the research work and the discussion of the results. Please show the pattern of microhardness measurements. Please report the results in terms of coefficients of S-N curve in order to make a quantitative evaluation of the statistical significance of the variations. Did you make any fatigue test on base material? It could be interesting to report them as for comparison.

Author Response

Please discuss the cost-effectiveness… (line 315-323)

Laser shock peening with a spot diameter SD = 2.0 mm and pulse density PD = 1600 laser pulses per square centimeter increased the fatigue life of the maraging precipitation hardened (MPH) steel specimens from range of 2 ÷ 4 ·104 cycles to range of 5 ÷ 9·105 cycles. Therefore, the number of fatigue cycles, necessary for fatigue crack initiation, increased by 25-times. In some cases we have stop the fatigue test after 107 cycles without crack initiation.

The implementation of LSP technique can increase fatigue resistance, by generating compressive residual stresses and inducing strain hardening in the most loaded surface areas, which consequently reduces the need for tool repair. So, when we succeed to increase tool maintenance interval (repair or change), the production cost is reduced.

Table 2: we have added the range of mechanical test results.

We have added Table 3: Research design: LSP process parameters and plan of measurements. (line 112).

The experimental setup of fatigue experiment is shematicaly presented in Figure 2 (line 140).

We have added Table 4: Fatigue loading parameters. (line 139).

LSP process parameters were correlated with the results of … (line 174-184)

To get an overview of the effects of laser pulse density and laser spot size on the surface integrity evaluated by roughness, hardness and residual stresses, the general factorial design was carried out. Laser parameters were examined using the analysis of variance (ANOVA) and response surface methodology (RSM), where the influence of individual factor is considered to be statistically significant when P<0.05. According to a statistical analysis ANOVA we found, that interactions of pulse density and spot diameter have a significant influence on the surface roughness, residual stresses, and hardness too (P<0.0001). We have tested several polynomial models on statistical characteristics (F-value, R2) to fit the response to measured values. We have found, according to F-values and R2 values, the most suitable model for the surface roughness, hardness, and residual stresses is the quadratic model. So according to mentioned measurements at laser spot diameter of 1.5, 2.0 and 2.5 mm, the results are presented as contour plots.

Why did you choose R=0.1…? (line 129-130)

We have chosen this stress ratio to provide and keep the upper specimen surface, where we expect crack initiation, in tensile condition during fatigue tests.

Additional information of numerical simulation of specimen stresses is provided (line 136-137)

Stress analysis was simulated with a software package SolidWorks according to finite elements method (FEM).

Did you observe any temperature increase…? (line 147-150)

We did not measure specimen temperature during fatigue testing, and we did not notice by touching any temperature increase in the specimens during and after the test. The CRACKTRONIC is a compact testing device, so possible heat generation in the specimen during fatigue loading could conduct to the resonant device.

In caption of Figure 2, we have added bending moment: (line 142)

M= 60 – 78 Nm.

We have added Figure 12: (line 267)

Micro-hardness measurements in depth for different pulse densities (PD=900; 1600; 2500 cm-2) and spot diameters (SD=1.5; 2.0; 2.5 mm)

Base metal was fatigue tested too. The fatigue test results were explained more in detail in text (line 300-305):

It was found out that maraging steel in as delivered state (maraging solution annealing, with hardness of 32-37 HRC, and Rm=1200 MPa) withstand the same number of fatigue cycles for crack initiation as maraging steel, which is precipitation hardened, when performing fatigue testing at bending moment of 60 Nm and bending stress of 833 MPa. When the bending stress is increased to 1082 MPa, precipitation hardened maraging steel (MPH) shows 43% longer crack initiation time (blue line in Figure 16).

I suggest to evaluate residual stresses with process… (line 350-357)

When choosing the best laser peening parameters for increasing fatigue life of a component, we must take into account the combination of the highest hardness and the highest compressive residual stresses at greater depth (in our case 1 mm). Laser spot size, in combination with laser power density and overlapping degree of laser pulses, affects the propagation of shock waves [26]. Small diameter shock waves attenuate with different rate into depth, as larger. In our case, we have obtained the best combination of surface quality and material properties at following LSP parameters: laser power density PPD = 8.9 GW cm-2, laser pulse density PD = 1600 cm-2, laser spot diameter SD = 2 mm, laser spot overlapping 87%.

Reviewer 3 Report

The manuscript entitled " Fatigue Properties of Maraging Steel after Laser 2 Peening"  deals with the influence of pulse density and spot size of laser shock peening process  on the surface integrity with fatigue resistance of X2NiCoMo18-9-5 maraging steel.

The content is very interesting and is worth to publish in the Metals journal.

However, before acceptance, some changes should be performed.

Please revise the manuscript according to the following comments:

Key words: should be in alphabetical order, KEY WORDS should not contain the same words that are within the title of the text.  Thus these should be changed appropriately

M&M

Paragraph 2.2. Laser shock peening. Please add the manufacturer name of the laser and also the laser parameters which were used (fluence, power density, application time).

2.3. Resonant fatigue tests Add a country and manufacturer name of the resonance testing machine.

Add manufacturer country and name of the profilometer.

Describe statistical analysis, and software was used to calculate a p-value.

Results

Add p values to the results.

The authors described in the results a pulse density and a power density as the same abbreviation of PD. This should be changed.

The unit of power density is in W/cm2. You should change the Results appropriately.

References

Abbreviated journal names should be added.

Author Response

Key words changed according the recommendation

We have added the manufacture name and country of the laser (also laser parameters that ere used), the resonant testing machine, and the profilometer.

Describe statistical analysis, and software…(line 174-180):

To get an overview of the effects of laser pulse density and laser spot size on the surface integrity evaluated by roughness, hardness and residual stresses, the general factorial design was carried out. Laser parameters were examined using the analysis of variance (ANOVA) and response surface methodology (RSM), where the influence of individual factor is considered to be statistically significant when P<0.05. According to a statistical analysis ANOVA we found, that interactions of pulse density and spot diameter have a significant influence on the surface roughness, residual stresses, and hardness too (P<0.0001).

P-values were added to the results.

Misnaming of pulse density (PD (cm-2)) and power density (PPD (W/cm2)) was corrected in the text. The results were checked and are OK.

Abbreviation of journal names are added.

Round 2

Reviewer 1 Report

none

Reviewer 3 Report

Thank you for the correction of the paper.

This manuscript is a resubmission of an earlier submission. The following is a list of the peer review reports and author responses from that submission.